# The Effect of Long Chain *n*-3 Fatty Acid Supplementation on Muscle Strength in Older Adults: A Systematic Review and Meta-Analysis

**DOI:** 10.3390/nu15163579

**Published:** 2023-08-14

**Authors:** Maha Timraz, Ahmad Binmahfoz, Terry J. Quinn, Emilie Combet, Stuart R. Gray

**Affiliations:** 1School of Cardiovascular and Metabolic Health, University of Glasgow, Glasgow G12 8TA, UK; m.timraz.1@research.gla.ca.uk (M.T.); a.binmahfoz.1@research.gla.ac.uk (A.B.); terry.quinn@glasgow.ac.uk (T.J.Q.); 2School of Medicine and Dentistry, University of Glasgow, Glasgow G31 2ER, UK; emilie.combetaspray@glasgow.ac.uk

**Keywords:** muscle strength, omega-3 polyunsaturated fatty acid, muscle mass, older adults

## Abstract

The main objective of the current study was to perform a systematic literature review with the purpose of exploring the impact of long-chain *n*-3 polyunsaturated fatty acid (LC*n*-3 PUFA) relative to control oil supplementation on muscle strength, with secondary outcomes of muscle mass and physical function in older individuals under conditions of habitual physical activity/exercise. The review protocol was registered with PROSPERO (CRD42021267011) and followed the guidelines outlined in the Preferred Reporting Items for Systematic Review and Meta-Analysis (PRISMA) statement. The search for relevant studies was performed utilizing databases such as PubMed, EMBASE, CINAHL, Scopus, Web of Science, and the Cochrane Central Register of Controlled Trials (CENTRAL) up to June 2023. Randomized controlled trials (RCTs) in older adults comparing the effects of LC*n*-3 PUFA with a control oil supplement on muscle strength were included. Five studies involving a total of 488 participants (348 females and 140 males) were identified that met the specified inclusion criteria and were included. Upon analyzing the collective data from these studies, it was observed that supplementation with LC*n*-3 PUFA did not have a significant impact on grip strength (standardized mean difference (SMD) 0.61, 95% confidence interval [−0.05, 1.27]; *p* = 0.07) in comparison to the control group. However, there was a considerable level of heterogeneity among the studies (*I*^2^ = 90%; *p* < 0.001). As secondary outcomes were only measured in a few studies, with significant heterogeneity in methods, meta-analyses of muscle mass and functional abilities were not performed. Papers with measures of knee extensor muscle mass as an outcome (*n* = 3) found increases with LC*n*-3 PUFA supplementation, but studies measuring whole body lean/muscle mass (*n* = 2) and functional abilities (*n* = 4) reported mixed results. With a limited number of studies, our data indicate that LC*n*-3 PUFA supplementation has no effect on muscle strength or functional abilities in older adults but may increase muscle mass, although, with only a few studies and considerable heterogeneity, further work is needed to confirm these findings.

## 1. Introduction

Muscle mass and function play a vital role in overall health and well-being, although this receives little research focus. Various conditions, such as sarcopenia, cachexia, and muscle disuse atrophy, can lead to muscle loss and deterioration in quality and function [1]. For example, as individuals reach the age of 35–40, muscle strength and mass gradually and progressively decline [2]. This process, termed sarcopenia, specifically refers to the age-related decrease in muscle strength and mass and is associated with several negative health outcomes. These include, but are not limited to, a decrease in quality of life, reduced functional capacity of muscles, physical disability, diminished quality of life, increased risk of mortality, and a higher likelihood of falls that may result in hospitalization [3]. Individuals with sarcopenia also incur substantially higher health and social care costs, including longer hospital stays and a greater need for residential living facilities [4,5]. In the United Kingdom alone, the additional health and social care expenses related to muscle weakness are estimated to reach GBP 2.5 billion per year [6].

Although the exact prevalence of sarcopenia remains uncertain, a recent meta-analysis involving a large number of individuals (*n* = 692,056, mean age: 68.5 years) estimated the global prevalence to be between 10% and 27% [7]. Furthermore, the proportion of older adults is expected to rise in many countries. For example, in the UK, there were 11.9 million (18% of the population) adults of pensionable age (67 years) in 2020, and this is projected to rise to 13.2 million (19% of the population) in 2030 and 15.2 million (21% of the population) in 2045. Similarly, again in the UK, in 2020, there were 1.7 million (2.5% of the population) adults aged 85 years or over, and this is projected to increase to 3.1 million (4.3% of the population) in 2045 [8]. Given this demographic shift, it is crucial to prioritize the development of treatments aimed at improving or slowing down the decline in muscle strength and mass among older adults.

Addressing the age-related decline in muscle strength and mass is of paramount importance. Effective medical interventions that can prevent or mitigate the negative consequences of muscle loss are, however, currently lacking [3]. As a result, there is a growing interest in exploring alternative approaches to maintain or enhance muscle strength and mass in older adults. Resistance training is the most effective method for increasing muscle strength and mass, even among nonagenarian women [9]. However, its effectiveness is reduced compared to younger individuals, primarily due to so-called “anabolic resistance” [10]. On top of that, low rates of participation significantly hinder the effectiveness of resistance exercise interventions targeting muscle strength and mass in older adults [11]. As a result, these interventions may not be as impactful as desired in terms of public health. To address this issue, modifying dietary habits has been proposed as a potential therapeutic approach to enhance muscle strength and mass in older individuals.

Among these approaches, adjusting protein and fatty acid consumption has been proposed as a potential therapeutic strategy [12]. One promising candidate in this regard is supplementation with the LC*n*-3 PUFA (long-chain omega-3 polyunsaturated fatty acids) Eicosapentaenoic acid (EPA) and Docosahexaenoic acid (DHA), found in marine sources such as oily fish and krill. There is a mixture of epidemiological, animal, cell culture, and acute human data that indicates the potential beneficial effect of EPA and DHA, but not other *n*-3 PUFA, such as alpha-linolenic acid (ALA), on muscle. For example, the consumption of fatty fish has a positive correlation with muscle strength among older populations, according to epidemiological data [13,14]. Cell culture and animal studies back up these findings [15,16]. Furthermore, 8 weeks of LC*n*-3 PUFA supplementation (4 g/day) increased muscle protein synthesis (MPS) during a hyperaminoacidaemic-hyperinsulinaemic clamp in humans [17]. However, overall the current evidence regarding the effects of LC*n*-3 PUFA on muscle strength and mass in older individuals is limited and inconsistent [18,19]. Additionally, the level of certainty in the available evidence is low [18]. Therefore, further research is needed to determine the efficacy of LC*n*-3 PUFA supplementation in order to address the age-related decline in muscle strength and mass effectively.

A recent scoping systematic review and meta-analysis, which included studies from a wide range of healthy and clinical populations, investigated whether this translates to increases in muscle strength and mass. Although studies were small and heterogeneity was high, the results indicated that LC*n*-3 PUFA supplementation can increase both muscle strength and mass [20]. There has also been a systematic review and meta-analysis restricted to older people [18], which included studies with exercise training alongside LC*n*-3 PUFA supplementation, which found an increase in muscle mass but not strength. Key studies, however, were unfortunately not included in this latter meta-analysis for muscle strength, and further studies have since been published. There is a clear need, therefore, for a robust and up-to-date meta-analysis investigating the impact LC*n*-3 PUFA on muscle strength and mass in older people in the absence of exercise training. The latter point is of interest to increase possible public health utility due to the previously mentioned issues with uptake and long-term adherence to exercise.

The objective of this study is to conduct a systematic literature review to investigate whether LC*n*-3 PUFA supplementation relative to control oil affects muscle strength, with secondary outcomes of muscle mass and functional performance in older adults under conditions of habitual physical activity/exercise.

## 2. The Materials and Methods

### 2.1. Data Sources and Searches

This systematic review adheres to the guidelines outlined in the Preferred Reporting Items for Systematic Review and Meta-Analysis (PRISMA) statement [21]. The study was registered with the International Prospective Register of Systematic Reviews (PROSPERO) in 2021 under the identification number CRD42021267011. A comprehensive search of the literature was conducted using several databases, including PubMed, EMBASE, CINAHL, Scopus, Web of Science, and the Cochrane Central Register of Controlled Trials (CENTRAL). Search terms are presented in Appendix A. Additionally, the reference lists of the included articles were examined to identify any additional articles that met the inclusion criteria outlined in the PICOS format described in Appendix A. We restricted our search to papers published in the English language, with no restriction on the publication period. The search was from the dates of inception until June 2023. Two researchers (MT and AB) independently screened the titles, abstracts, and full texts of the identified studies to determine their eligibility. In case of disagreements, a third researcher (SRG) was consulted to resolve conflicts.

### 2.2. Selection Criteria

The study employed specific criteria for inclusion, namely randomized controlled trials that investigated the effects of LC*n*-3 PUFA (EPA and DHA) supplementation (with no other changes to habitual diet or physical activity) on muscle strength in older individuals. The mean age of the participants in the sample studies was 65 years and older. The primary focus of the studies was to assess muscle strength as the main outcome, while secondary outcomes encompassed muscle mass and muscle function, including functional tests such as the time-up-and-go test. Studies that did not provide primary data, such as abstracts, meta-analyses, and reviews, were considered ineligible for inclusion. Appendix A presents the detailed inclusion and exclusion criteria with respect to this study’s PICOS framework.

### 2.3. Data Extraction and Quality Assessment

Data extraction was independently performed by two investigators (MT and AB) using a pre-specified data collection form. The following information was extracted during the data extraction process: authors’ names, publication year, study design, sample size, mean age, gender, population characteristics, duration of follow-up, intervention period, details of exercise, type of LC*n*-3 PUFA, dosage (in grams per day), and outcomes related to muscle mass, muscle strength, and muscle function. Two investigators (MT and AB) utilized the Cochrane Collaboration risk-of-bias tool [22] to evaluate the quality of evidence. In the event of any discrepancies, a third reviewer (SRG) was involved in the discussion to reach a consensus. The level of bias was categorized as high, low, or unclear based on seven criteria: random sequence generation, allocation concealment, blinding of participants and personnel, blinding of outcome assessment, incomplete outcome data, selective reporting, and other potential sources of bias. The strength of the recommendations and the quality of the evidence were assessed using the Grading of Recommendations Assessment, Development, and Evaluation (GRADE) methodology. Judgments were made based on criteria such as the risk of bias, inconsistency, imprecision, indirectness, and publication bias, and the quality of evidence was classified as high, moderate, low, or very low.

### 2.4. Data Synthesis and Statistical Analysis

The meta-analyses were carried out utilizing Review Manager Version 5.3 (RevMan) software. The effect size was estimated as standardized mean differences (SMDs) with 95% confidence intervals (CIs) and employing a random-effects model [17]. The mean changes in scores (final–baseline) and SDs were used. When these data were not available, the authors were contacted, and any missing change SDs or median to mean conversions were carried out using methods described previously [23]. Statistical heterogeneity was evaluated by performing a chi-squared test and the calculation of the *I*^2^ statistic.

## 3. Results

### 3.1. Study Identification

Our initial search strategy identified 2931 studies, out of which 1882 remained after removing duplicates (Figure 1). Following the screening of titles and abstracts, 26 full-text studies underwent a thorough review for eligibility, resulting in the inclusion of 5 studies. (Table 1). There were 488 participants in these articles (348 females and 140 males). All the studies included in the analysis were conducted as randomized controlled trials.

### 3.2. Study Characteristics

Table 1 summarises the main characteristics of the five studies that were included in the analysis. Among these studies, two were conducted in the United States, one in the United Kingdom, one in Canada, and one in China. The average age of participants varied between 66 and 75 years old. The duration of interventions was between 12 and 24 weeks. The five RCTs included healthy, community-dwelling older adults. Two of the five studies included only women and one only included postmenopausal women particularly. The rest included both males and females. The studies included the main outcomes of muscle mass, muscle strength, and muscle function [23,24,25,26,27]. nutrients-15-03579-t001_Table 1Table 1Studies investigating the impact of omega-3 supplementation on measures of muscle strength and muscle function in older adults.AuthorCountryDesignSampleInterventionMain Effects of LC*n*-3 PUFA Relative to PlaceboAlkhedhairi et al. [27].UKDouble-blind *n* = 94; *n* = 53 female, *n* = 41 male; age = 71 ± 5 years.(4 g/day; of Krill oil 772 mg/d EPA and 384 mg/day DHA) or placebo (4 g/day mixed vegetable oil) for 6 months.Increase in grip strength (10.9%), leg strength (9.3%), and vastus lateralis muscle thickness (3.5%). No effect on the short-performance physical battery test or whole-body bioelectrical impedance measured muscle mass.Hutchins-Wiese et al. [26].USADouble-blind *n* = all females; 126; age = 75 ± 6 years.(2 capsules Fish oil, 1.2 g/day EPA and DHA) or placebo (2 capsules olive oil, 1.8 g/day olive oil) for 24 weeks.No effect on grip strength or repeated chair rise test. Increased walking speed (3%) (*p* = 0.038).Logan [25]. CanadaSingle-blind *n* = all female; 24; age = 66 ± 1 years.(5 g/day Fish oil (2 g/day EPA and 1 g/day DHA)) or placebo (3 g/day olive oil) for 12 weeks.No effect on grip strength or the 30-s sit-to-stand test. Increased whole-body bioelectrical impedance measured lean mass (4%) and TUG test (7%).Smith et al. [24].USADouble-blind *n* = 44; male = 15 and female = 29; age = 69 ± 6 years.(4 × 1 g pills/day of Fish oil providing 1.86 g/day EPA and 1.5 g/day DHA) or placebo (4 × 1 g pills/day of corn oil) for 6 months.Increased thigh muscle volume (3.6%), handgrip strength (2.3 kg), and 1-RM muscle strength (4.0%).Average isokinetic power tended to be increased (5.6%) (*p* = 0.075).Dengfeng Xu [28].ChinaDouble-blind *n* = 200; female, *n* = 116; male, *n* = 84; age = 67 ± 5 years.(4 g/day; 1.34 g of Fish oil/d EPA and 1.07 g/day DHA) or placebo (4 g/day corn oil) for 6 months.Increased thigh muscle volume (3.66 cm), handgrip strength (4.91 kg), and Timed Up and Go strength (1.85 s).


### 3.3. Intervention and Comparators

The supplementation regimens employed in the studies are outlined in Table 1. Among the included studies, four of them [24,25,26,28] utilized either fish oil or LC*n*-3 PUFA derived from fish oil as the supplementation, while one study used krill oil [27]. In terms of comparators, two studies used olive oil [25,26], one used mixed vegetable oil [27], and two used corn oil [24,28]. These details provide an overview of the different supplementation and comparator options employed across the studies.

### 3.4. Risk of Bias

Appendix A provides an overview of the risk of bias observed in the studies. Overall, the studies demonstrated a low risk of bias in several domains, including random sequence generation, blinding of participants and personnel, allocation concealment, and selective reporting. Two studies [24,25] were deemed to have a high risk of bias regarding allocation concealment, while one study [26] had an unclear risk of bias in this domain. In terms of the blinding of outcome assessment, three studies [24,25,28] were classified as having a high risk of bias. One of the studies showed an unclear risk of bias for incomplete data outcomes [26], and five studies had other potential biases [24,25,26,27,28] (Appendix A). Studies with a high risk of bias in the domain of “measurement of the outcome” were categorized as such because there was no blinding of investigators. Because it could not be determined whether data were analyzed according to a pre-specified plan or registered before recruitment, studies with some concerns were rated as such in the “selection of reported results” domain (Appendix A).

### 3.5. The Effects of LCn-3 PUFA on Muscle Mass and Function

In terms of muscle mass, four studies involving 362 participants (140 males and 222 women) reported the effects of LC*n*-3 PUFA in older adults compared with a control group [24,25,27,28]. One study reported no significant impact of LC*n*-3 PUFA on muscle mass, estimated by bioelectrical impedance, but did report an increase in vastus lateralis muscle thickness assessed through ultrasound measurements [27]. One study found a 3.6% increase in thigh muscle volume, measured by MRI [24], and another reported an increase in whole-body lean mass, measured by bioelectrical impedance [25]. One study found significant increases in both skeletal muscle mass (*p* < 0.001, interaction time × group effect) and appendicular skeletal muscle mass (*p* < 0.001, interaction time × group effect) [28].

In terms of functional abilities, four studies involving 444 participants (125 male and 319 female) reported the effects of LC*n*-3 PUFA in older adults [25,26,27,28]. One study indicated no influence of LC*n*-3 PUFA on muscle function, as evaluated through the short-performance physical battery test [27]. In one study, no significant effect of LC*n*-3 PUFA on repeated chair rise performance was observed [26]. However, in another study, data demonstrated a 3% increase in walking speed (*p* = 0.038) associated with the performance of the timed-up-and-go (TUG) test [25]. Furthermore, a separate study demonstrated a significant reduction in the TUG time (*p* < 0.001), indicating an improvement in mobility [28].

### 3.6. The Effects of n-3 Fatty Acids on Hand Grip Strength—Meta-Analysis

Figure 2 depicts the combined effects of LC*n*-3 PUFA supplementation on hand grip strength. Five studies [24,25,26,27,28] involving 488 participants (n = 140 males, 348 females) were included. The analysis revealed no statistically significant impact of LC*n*-3 PUFA supplementation (SMD 0.61, [−0.05, 1.27], *p* = 0.07) on hand grip strength compared to the control group. However, a considerable level of heterogeneity was observed (*I*^2^ = 90%, *p* < 0.001) among the included studies, suggesting variations in the results across the studies. Our GRADE (Appendix A) analysis revealed our certainty in these results was very low. 

## 4. Discussion

In this systematic review, we aimed to investigate the effects of LC*n*-3 PUFA (long-chain omega-3 polyunsaturated fatty acids) supplementation on various aspects of muscle health, including muscle strength, muscle mass, and muscle function, in older adults without underlying health conditions, with a meta-analysis performed for the primary outcome of muscle strength. By analyzing and synthesizing the findings from five randomized controlled trials that involved a total of 488 healthy older adults, we were able to assess the overall impact of LC*n*-3 PUFA supplementation on these muscle-related outcomes. The results of our meta-analysis indicate that LC*n*-3 PUFA supplementation did not have a significant impact on handgrip strength in healthy older adults. This suggests that LC*n*-3 PUFA supplementation may not be sufficient to improve handgrip strength in this population. The current data are based on only five studies, and our analysis of statistical heterogeneity indicates that there is considerable heterogeneity, meaning the results of the individual studies are not consistent and should be interpreted with caution. These issues limit our ability to draw firm conclusions, and further work is needed to establish whether LC*n*-3 PUFA is a potential strategy to counteract the age-related decline in muscle strength. In addition to the meta-analysis, we conducted a narrative synthesis to provide a comprehensive overview of the available evidence. The narrative synthesis revealed that LC*n*-3 PUFA supplementation may have a positive influence on muscle mass, as there were indications of increased muscle mass in three studies included in our analysis. However, when it came to muscle function, the results were more mixed, and we did not find a clear beneficial effect of LC*n*-3 PUFA supplementation on muscle function outcomes.

Given the limitations of the current evidence, including the limited data and high heterogeneity among the studies, further research is needed to provide a more definitive understanding of the effects of LC*n*-3 PUFA supplementation on muscle strength and mass in older adults. Studies included in the current review had durations of 12 weeks to 6 months and provided doses of ~1–3 g/day LC*n*-3 PUFA, with insufficient current data to investigate whether dose and/or duration of supplementation can influence the effects of LC*n*-3 PUFA on muscle. Future studies should aim to address these gaps in the literature. Ultimately, it remains uncertain whether LC*n*-3 PUFA supplementation can effectively mitigate the age-related decline in muscle strength and mass associated with aging. Therefore, more robust, well-designed studies are warranted to determine the potential utility of LC*n*-3 PUFA supplementation as a viable approach to promoting muscle health in older adults without underlying health conditions.

Muscle strength and mass naturally decline as individuals age, typically starting around 35–40 years, and can eventually lead to the development of sarcopenia [29]. Unfortunately, there are currently no effective and safe medical treatments available for either the prevention or treatment of this condition [30]. However, emerging research suggests that nutrition may offer a potentially effective approach to delay the age-related decline in muscle mass and function among older individuals [12], with LC*n*-3 PUFA emerging as a strong candidate. The current data indicate that LC*n*-3 PUFA does not have any significant impact on grip strength in older adults. This finding aligns with previous older meta-analyses that had broader inclusion criteria [18,19]. For example, in [18], studies with and without exercise and/or multi-nutrient supplements were included, with no effect of LC*n*-3 PUFA on grip strength (on the basis of 3 studies with 97 participants). Similarly, in [19], no effect on grip strength was noted in people 55 years or older (on the basis of 7 studies with 680 participants). In our data, there appears to be a tentative trend suggesting a potential beneficial effect of LC*n*-3 PUFA on grip strength, and some studies have indicated its potential benefits on lower body muscle strength. While grip strength is commonly used in the diagnostic criteria for sarcopenia due to its simplicity [31], it can be argued that lower body muscle strength holds greater practical significance for older individuals in terms of maintaining mobility and independent living.

Based on our assessment using the GRADE framework, the level of certainty in the evidence derived from the present study was determined to be low. Therefore, further research is necessary to explore the effects of LC*n*-3 PUFA on grip strength in older adults. The exact mechanisms through which LC*n*-3 PUFA might influence muscle strength are not fully understood. However, potential explanations may involve improvements in neuromuscular function, enhanced blood flow for better nutrient delivery, increased mitochondrial content and function, and alterations to the extracellular matrix to facilitate enhanced force transmission [32].

Our analysis of the effects of LC*n*-3 PUFA on muscle function yielded varied results in the narrative synthesis. However, when considering the overall findings, it is suggested that there is no discernible beneficial effect. Our review of the literature on functional abilities had mixed results, with some finding benefits on walking speed [26] and TUG (timed up and go test) [25,28], and others finding no effect on chair rise tests [25,26] or the short-performance physical battery test [27]. This is perhaps not surprising, as in the studies included in the current analysis, participants were all healthy and so had no functional limitations, a finding exemplified in the recent Do-HEALTH study, which was a 2 × 2 × 2 factorial design study investigating the effects of vitamin D, a home exercise program, and LC*n*-3 PUFA on a range of clinical outcomes, including the (SPPB) short-performance physical battery test [33]. In this study, involving 2157 older adults who received 3 years of LC*n*-3 PUFA supplementation, no significant impact on the short-performance physical battery test score was observed. The participants, however, had baseline scores of 11–12 out of 12 and were highly physically active. Whether LC*n*-3 PUFA can influence muscle function in older adults with functional limitations remains to be investigated in further studies in people with pre-existing limitations in functional capacity.

When looking at muscle mass as an outcome, our data indicate a beneficial effect of LC*n*-3 PUFA on muscle mass, which agrees with the previous, more broad meta-analyses on this topic [17,19]. In the current review, it was found that MRI-measured thigh muscle volume [23], ultrasound-measured vastus lateralis muscle thickness [27], and whole-body bioelectrical impedance-evaluated lean mass, skeletal muscle mass, body fat mass, and fat-free mass [25,28], were increased following LC*n*-3 PUFA supplementation. No effect on whole-body bioelectrical impedance-measured muscle mass was seen in one study, although increases in muscle thickness were seen [26]. The potential mechanisms by which LC*n*-3 PUFA may enhance muscle mass are associated with muscle protein metabolism, specifically the equilibrium between muscle protein synthesis (MPS) and muscle protein breakdown (MPB). To date, there have been no studies that have directly measured muscle protein breakdown (MPB) in response to LC*n*-3 PUFA supplementation. However, research has demonstrated that LC*n*-3 PUFA supplementation can elevate muscle protein synthesis (MPS) during a hyperinsulinemic-hyperaminoacidemic clamp, as observed in both young and older adults [17,34]. These increases in MPS have been accompanied by elevated levels of signaling proteins (p70s6k and mTOR) associated with translation activation [17,35]. These increases in muscle mass may contribute to an increase in muscle strength due to LC*n*-3 PUFA, but there are also potential mass-independent mechanisms through which strength can be increased. DHA is an essential constituent of membrane phospholipids which are crucial for numerous neural functions such as receptor affinity and modulation of signal transduction [36]. In younger adults, there is some evidence that LC*n*-3 PUFA supplementation can increase surface electromyography (sEMG) levels and muscle power in young male athletes [37]. There is less evidence in older adults, but it has been shown that LC*n*-3 PUFA can increase the M-wave, an indirect measure of fiber membrane excitability, and has shown notable decreases in clinical myopathies [38] in older adults. Other potential mechanisms include improvements in mitochondrial content and function, blood supply and extracellular matrix content, composition and architecture, and the anti-inflammatory effects of LC*n*-3 PUFA [32].

## 5. Conclusions

The findings of our meta-analyses should be interpreted with caution due to several notable limitations. These include the limited number of studies and participants involved, as well as the presence of substantial heterogeneity across the included studies. This makes our conclusions tentative. Supplementation with LC*n*-3 PUFA seems to have a positive impact on muscle mass; however, its effect on muscle strength and function remains inconclusive. Therefore, additional research is necessary to determine if LC*n*-3 PUFA supplementation has utility as a strategy for preventing or treating age-related declines in muscle strength, mass, and functional abilities. Additionally, optimal dosing strategies should be investigated.

## Figures and Tables

**Figure 1 nutrients-15-03579-f001:**
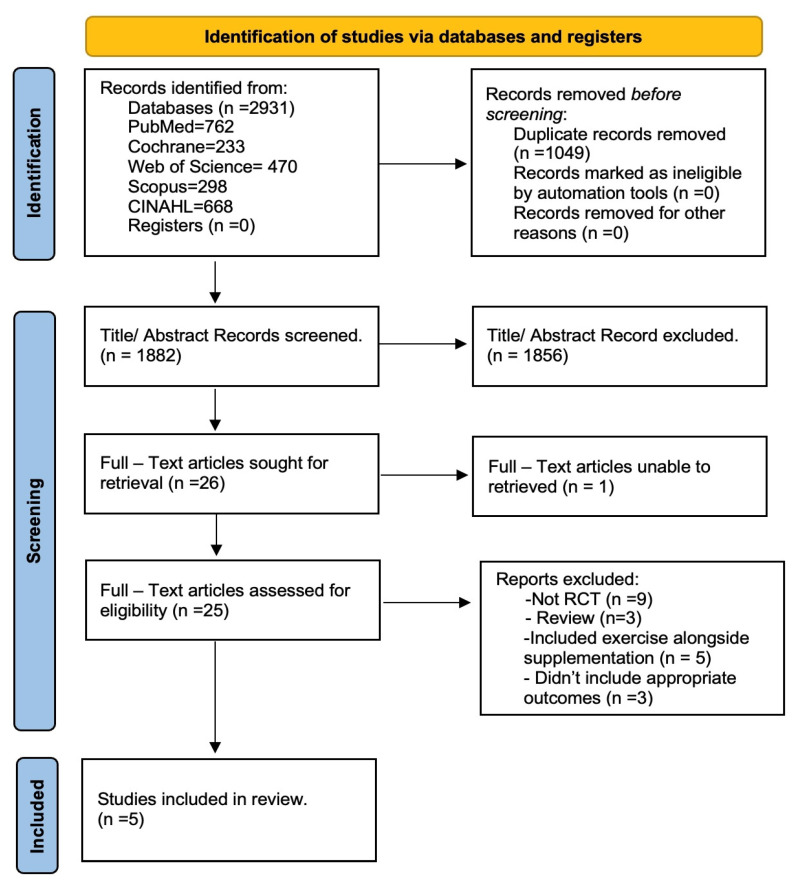
PRISMA diagram: flow chart of the study selection process.

**Figure 2 nutrients-15-03579-f002:**
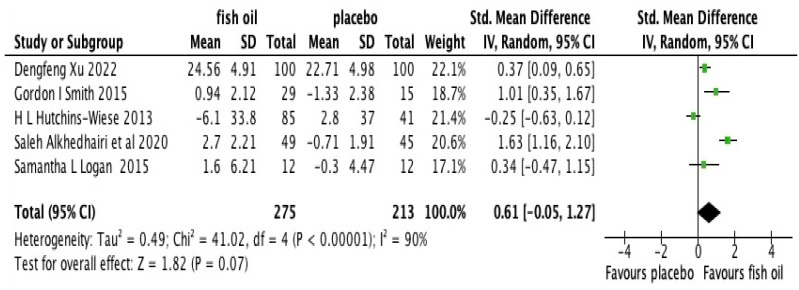
A meta-analysis on the effects of n-3 fatty acids on hand grip strength in older adults [24,25,26,27,28].

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
