# Peer review of "The Effect of Long Chain *n*-3 Fatty Acid Supplementation on Muscle Strength in Older Adults: A Systematic Review and Meta-Analysis"

_nutrients, 2023, doi:10.3390/nu15163579_

Round 1
Reviewer 1 Report
The manuscript deals with themes that are current and bear importance when considering dietary/lifestyle interventions. The text reads well.
Abstract and main text: what means "up to June 2023"? Were the authors able to perform complex analyses days before submission? It seems hardly possible.
Abstract and Meta-analysis: since only a few studies are considered for evaluation, the term "heterogeneity" should be elaborated, not only by providing a coefficient but also substantively, as it directly impacts the reasoning.
Abstract and Discussion: if analysed studies report mixed results on something, one may not state a lack of effect on that something. Please consider revising these formulations.
Introduction: current epidemiological numbers and projections should be provided. An appropriate citation would be welcome.
Authors state that reference [14] reported at least partially different results from the current study. Since it is the only citation of a similar kind, one might consider a more detailed discussion of the differences in obtained results.
A major point results from the inclusion/exclusion criteria used, particularly the question of age. Section 2.2 states "older adults aged 65 and older". Table 1 summarises the analysed studies and, practically, none of them meets this age criterion. Furthermore, Supplementary Table 2 mentions the inclusion criterion "healthy adults with MEAN sample age 65 years" whereas the exclusion criterion states "Participants who were pregnant or had major disease". Which diseases are thus non-major and constitute a healthy subject? Moreover, do the authors really expect pregnant women in the studied age group? In "other" the mention of "human subjects" seems redundant. These parts of the manuscript deserve a solid rework.
In Results, mentioning RCT as an inclusion criterion is also redundant.
In Discussion, the first sentence is not correct: the meta-analysis was not done for muscle mass and function. The sentence "Currently, there is no effective medical treatment..." cites a publication from 2015. A more current citation is needed here.
In general, it is common knowledge that dietary interventions without a component of physical activity have no or a limited effect on muscle strength and function. The manuscript would benefit from being more specific on the impact of LCn-3 PUFA on these parameters.
Author Response
Reviewer1 Responses:
- Abstract and main text: what means "up to June 2023"? Were the authors able to perform complex analyses days before submission? It seems hardly possible.
Response 1:
Yes, we ran an updated search in June 2023 and screen the new articles for any that were eligible for inclusion in the current systematic review/meta-analysis.
- Abstract and Meta-analysis: since only a few studies are considered for evaluation, the term "heterogeneity" should be elaborated, not only by providing a coefficient but also substantively, as it directly impacts the reasoning.
Response 2:
We have added details of our statistical test for heterogeneity to the methods section and elaborated in our discussion on how this can limit our interpretation of the data.
- Abstract and Discussion: if analysed studies report mixed results on something, one may not state a lack of effect on that something. Please consider revising these formulations.
Response 3:
We are in agreement and have altered our conclusions in abstract to reflect our uncertainty. Similarly at the beginning and end of the discussion we have softened our conclusions we can draw from the current data.
- Introduction: current epidemiological numbers and projections should be provided. An appropriate citation would be welcome.
Response 4:
We assume that this in relation to the data on the ageing population and we have added in updated statistics and a reference for these.
- Authors state that reference [14] reported at least partially different results from the current study. Since it is the only citation of a similar kind, one might consider a more detailed discussion of the differences in obtained results.
Response 5:
In the current version this is reference 18 and we have elaborated on this and reference 19 in the discussion.
- A major point results from the inclusion/exclusion criteria used, particularly the question of age. Section 2.2 states "older adults aged 65 and older". Table 1 summarises the analysed studies and, practically, none of them meets this age criterion. Furthermore, Supplementary Table 2 mentions the inclusion criterion "healthy adults with MEAN sample age 65 years" whereas the exclusion criterion states "Participants who were pregnant or had major disease". Which diseases are thus non-major and constitute a healthy subject? Moreover, do the authors really expect pregnant women in the studied age group? In "other" the mention of "human subjects" seems redundant. These parts of the manuscript deserve a solid rework.
Response 6:
We have edited the table to make clear that we included studies where the sample had a mean age of 65 years or older. We have deleted human subjects and reference to pregnancy. We have also clarified what we constitute as major disease.
- In Results, mentioning RCT as an inclusion criterion is also redundant.
Response 7:
We have altered the results accordingly.
- In Discussion, the first sentence is not correct: the meta-analysis was not done for muscle mass and function. The sentence "Currently, there is no effective medical treatment..." cites a publication from 2015. A more current citation is needed here.
Response8:
- We have altered the first sentence of the discussion.
- We have update this citation to a more recent one.
- In general, it is common knowledge that dietary interventions without a component of physical activity have no or a limited effect on muscle strength and function. The manuscript would benefit from being more specific on the impact of LCn-3 PUFA on these parameters.
Response 9:
We would disagree with this statement. We currently do not know if any nutritional strategies can be effect in increasing muscle strength and function in older adults, in the absence of exercise. As data indicates LCn-3 PUFA may increase muscle strength and function in older adults in the absence of exercise we have performed a meta-analysis on this topic, something that was missing from the current literature and for which we made a rationale in the introduction.
Reviewer 2 Report
This study shows that LCn-3 PUFA has no significant effect on the muscle strength of older adults, and the questions are as follows.
1. This study investigated the effects of LCn-3 PUFA on muscle strength. What is the relationship between diet and muscle strength? Why choose long-chain n-3 fatty acids for research and analysis?
2. LCn-3 PUFA, not only EPA and DHA but also linolenic acid, why was this fatty acid not included in the analysis of this study?
3. Please explain the possible mechanism between LCn-3 PUFA and muscle strength.
4. Did the studies include considering the lifestyle of the participants?
5. The heterogeneity is high among the included studies. The effect of LCn-3 PUFA on muscle strength could obtain intangible information from the systematic and meta-analysis.
6. Please discuss the effects of LCn-3 PUFA supplementation on dosage and duration in these studies.
7. Please introduce the dietary sources of LCn-3 PUFA, and re-set the keywords (such as linolenic acid, linseed oil, and perilla oil) for meta-analysis.
Author Response
Reviewer 2 Responses:
- Comment: This study investigated the effects of LCn-3 PUFA on muscle strength. What is the relationship between diet and muscle strength? Why choose long-chain n-3 fatty acids for research and analysis?
Response 1:
This is a very broad comment and currently we do not know what the relationship between “diet” and strength is. This was part of the rationale for this study. We choose LCn-3 PUFA due to the literature cited in the introduction that shows that epidemiological, animal, cell culture and acute human physiology studies indicated that there may be benefits of LCn-3 PUFA for muscle strength.
- Comment: LCn-3 PUFA, not only EPA and DHA but also linolenic acid, why was this fatty acid not included in the analysis of this study?
Response 2:
We have chosen to restrict our analysis to EPA and DHA rich marine supplements due to the aforementioned epidemiological, animal, cell culture and acute human physiology studies that indicated the potential relationship between the LCn-3 PUFAs EPA and DHA, but not others, on muscle mass and strength. We have tried to make this clear in the introduction and will restrict the review to EPA and DHA rich supplements.
- Comment: Please explain the possible mechanism between LCn-3 PUFA and muscle strength.
Response 3:
We have added more detail to the discussion on the potential mechanisms, although this is also an area where further work is needed.
- Comment: Did the studies include considering the lifestyle of the participants?
Response 4:
We excluded studies, as detailed in supplementary table 2, with co-interventions such as exercise or other nutritional supplements. The studies included has very little characterisation of the broader lifestyle of participants unfortunately.
- Comment: The heterogeneity is high among the included studies. The effect of LCn-3 PUFA on muscle strength could obtain intangible information from the systematic and meta-analysis.
Response 5:
We have tried to temper our conclusions and re-iterate the limitation to the current data, including the high heterogeneity.
- Please discuss the effects of LCn-3 PUFA supplementation on dosage and duration in these studies.
Response 6:
This information has been detailed in table 1 and the results, and we have elaborated on this in the discussion.
- Please introduce the dietary sources of LCn-3 PUFA, and re-set the keywords (such as linolenic acid, linseed oil, and perilla oil) for meta-analysis.
Response 7:
As detailed above our rationale and supporting data indicate potential effects of the LCn-3 PUFA EPA and DHA, but not other fatty acids, on muscle and so we have restricted our search to these supplements.
Round 2
Reviewer 1 Report
The manuscript got notably better after the first review round. There are, however still a few issues that can be classified as minor:
Figure 1: If 2 articles, out of 26 were, excluded due to unavailability, the number of 25 remaining is incorrect.
Table 1: there is still inconsistency with reporting the age, please unify.
Furthermore, with implications for Discussion, both studies that delivered no evidence of an effect on hand grip strength were performed on female subjects, which should be discussed. Particularly so, in the wake of the formulation "The results of our meta-analysis indicate that LCn-3 PUFA supplementation did not have a significant impact on handgrip strength in healthy older adults" in Discussion, which appears incorrect.
Please check Figure 1 for formulations used.
Author Response
1-Figure 1: If 2 articles, out of 26 were, excluded due to unavailability,
the number of 25 remaining is incorrect.
Response 1:
[As you can see in the corrected figure, it is 1 article, out of 26 not 2 articles. So, the number of 25 remaining there. Sorry for this error].
2- Table 1: there is still inconsistency with reporting the age, please unify.
Response 2:
[This has been corrected in the table, so all data are mean SD].
3- Furthermore, with implications for Discussion, both studies that
delivered no evidence of an effect on hand grip strength were performed
on female subjects, which should be discussed. Particularly so, in the
wake of the formulation "The results of our meta-analysis indicate that
LCn-3 PUFA supplementation did not have a significant impact on handgrip
strength in healthy older adults" in Discussion, which appears incorrect."
Response 3:
[We would disagree with this comment as we are not certain that the lack of effect in these studies is because they are only women included. We think we have made a tentative conclusion, based on current data, and highlighted that further work is needed to confirm this. We have also looked at our data and the data of Smith et al and if anything there is a larger effect of omega-3 in women compared to men. This is unpublished data.].